# Prediction Models for Public Health Containment Measures on COVID-19 Using Artificial Intelligence and Machine Learning: A Systematic Review

**DOI:** 10.3390/ijerph18094499

**Published:** 2021-04-23

**Authors:** Anil Babu Payedimarri, Diego Concina, Luigi Portinale, Massimo Canonico, Deborah Seys, Kris Vanhaecht, Massimiliano Panella

**Affiliations:** 1Department of Translational Medicine (DIMET), Università del Piemonte Orientale, 28100 Novara, Italy; diego.concina@uniupo.it (D.C.); massimiliano.panella@med.uniupo.it (M.P.); 2Department of Science and Technological Innovation (DISIT) Università del Piemonte Orientale, 15121 Alessandria, Italy; luigi.portinale@uniupo.it (L.P.); massimo.canonico@uniupo.it (M.C.); 3Leuven Institute for Healthcare Policy, Department of Public Health and Primary Care, KU Leuven, 3000 Leuven, Belgium; deborah.seys@kuleuven.be (D.S.); kris.vanhaecht@kuleuven.be (K.V.); 4Department of Quality Management, University Hospitals Leuven, University of Leuven, 3000 Leuven, Belgium

**Keywords:** artificial intelligence, machine learning, COVID-19, public health interventions, prediction models, epidemic, pandemic, severe acute respiratory syndrome coronavirus-2

## Abstract

Artificial Intelligence (AI) and Machine Learning (ML) have expanded their utilization in different fields of medicine. During the SARS-CoV-2 outbreak, AI and ML were also applied for the evaluation and/or implementation of public health interventions aimed to flatten the epidemiological curve. This systematic review aims to evaluate the effectiveness of the use of AI and ML when applied to public health interventions to contain the spread of SARS-CoV-2. Our findings showed that quarantine should be the best strategy for containing COVID-19. Nationwide lockdown also showed positive impact, whereas social distancing should be considered to be effective only in combination with other interventions including the closure of schools and commercial activities and the limitation of public transportation. Our findings also showed that all the interventions should be initiated early in the pandemic and continued for a sustained period. Despite the study limitation, we concluded that AI and ML could be of help for policy makers to define the strategies for containing the COVID-19 pandemic.

## 1. Introduction

During the last five years, the use of Artificial Intelligence (AI) and Machine Learning (ML) rapidly increased its applications in various areas of medicine [1,2]. In particular, during the SARS-CoV-2 outbreak, AI and ML were shown to be effective in improving diagnostic and prognostic processes of COVID-19, although there were limitations due to potential biases relating to the quality of reporting [3]. AI and ML were also applied to public health issues related to COVID-19. This included the identification of clinical and social factors associated with the risk of COVID-19 infections and deaths [4,5,6], the development of spatial risk maps [5], the prediction of the trends and peak of the epidemic [7,8], and finally, the development of vaccination strategies [9]. For optimizing protection and preventing the spread of COVID-19, several activities need to be implemented, such as the identification of suspicious events, large-scale screening, tracking, associations with experimental treatments, pneumonia screening, data and knowledge collection and integration using the Internet of Intelligent Things (IIoT), resource distribution, robotics for medical quarantine, forecasts, and modeling and simulation [10,11].

In the actual phase of the epidemic, governments are using public health measures such as lockdown, social distancing, and school closures, etc., to contain the spread of the virus. The effectiveness of such strategies is mainly based on theoretical assumptions [12]. Moreover, epidemiological models such as Susceptible–Exposed–Infectious–Recovery (SEIR), stochastic transmission models, etc., that have traditionally been used to study and predict dynamics and possible contagion scenarios [13,14,15] also had limited application to public health interventions for the COVID-19 pandemic.

We think that AI and ML could be a good opportunity to address such issues and to help policy makers in strengthening the selection of the most appropriate public health measures against COVID-19 [10,16,17]. Therefore, we decided to conduct a systematic review to evaluate the effectiveness of AI and ML to guide the implementation of public health interventions aimed to contain the SARS-CoV-2 pandemic.

## 2. Materials and Methods

### 2.1. Search Strategy

We searched Nursing Reference Center Plus, CHINAHL, Scopus, PubMed and Living Evidence [18] (contains studies continuously updated on COVID-19 that are published on PubMed and Embase through Ovid, bioRxiv, and medRxiv and is continuously updated) on 1 February 2020. The search strings are shown in the online Appendix A. All studies were considered, irrespective of their languages or publication status (preprint; updates of preprints are included and reassessed after publication in journals).

### 2.2. Inclusion Criteria

We included all the studies that used AI and/or ML to develop or validate a public health intervention and their possible outcomes. Titles, abstracts, and full texts of articles were screened for eligibility in duplicate by independent reviewers (A.B.P., D.C., D.S.) and discrepancies were resolved through discussion (M.P., K.V.).

### 2.3. Data Extraction

Two reviewers (A.B.P., D.C.) independently extracted data from all the included articles by using a predefined data extraction form which contained the following variables: title, setting, type of model (new or existing), typology of data, outbreak phase, outcome, intervention type, intervention description and results of the study. Any discrepancies in the data extraction were discussed between reviewers, and the conflicts were resolved by M.P. and K.V. We adapted the PRISMA statement (preferred reporting items for systematic reviews and meta-analyses) for the articles’ selection [19]. A qualitative synthesis was performed for the included studies.

### 2.4. Definition of the Interventions

A quarantine is the isolation of the population exposed to COVID-19. A full lockdown is the containment strategy able to minimize contact between individuals. This includes shutting down the government departments, firms, schools, social and leisure facilities, and transportation services, keeping only essential services open, such as health, safety, and basic utilities [20]. A partial lockdown is the mitigation of lockdown according to the spatial (area) risk of the spread of the disease; in this approach, the different intensity of lockdown restrictions in one area is continuously adapted according to the change of parameters of the risk of diffusion of the disease (low, moderate, and high) [21]. Social distancing is the maintenance of 2 m of space between individuals and others outside one’s household (this includes avoiding groups, large gatherings) [20].

## 3. Results

We found and retrieved 3041 articles. After the removal of 14 duplicate records, 3027 articles were retained for screening. After the screening, 2943 records were excluded, and 84 full-text articles were assessed for eligibility. After the assessment, 76 full-text articles were excluded, and eight studies met the inclusion criteria and were included in the qualitative synthesis (Figure 1).

In Table 1, the study characteristics are described. Six studies used data from a single nation. Two studies used data from multiple countries.

Five studies used existing models. Two studies used a susceptible–infected–recovered (SIR) model to evaluate the spread of the epidemic. The first one performed a cross-national data analysis (26 countries) to evaluate the efficacy of social distancing interventions and analyzed the transmission rates of the disease over the course of 5 weeks [22]. The second one built a SEIRD model (MATLAB R2017a) based on the movement of people across regions, revealing the effects of three public health measures on the control of the epidemic [23].

Pasayat et al. combined mathematical (Exponential Growth) and ML (Linear Regression) models to predict the rates of COVID-19 cases in India with concern to lockdown intervention [24]. Marini et al. used EnerPol, a holistic agent-based model, to predict the growth of the epidemic according to containment strategy in Switzerland [25]. Qiu et al. adapted an empirical model that examined the role of various socioeconomic mediating factors, including public health measures encouraging social distancing in local communities, in reducing COVID-19 transmission [26].

Three studies used new models. Wang et al. proposed a new mechanistic model describing the transmission of COVID-19 in the United States [27]. Kumar et al. proposed a prospective methodology using TOPSIS (Technique for Order of Preference by Similarity to Ideal Solution). This consisted of the multi-criteria decision-making technique able to measure the spatial footprint of COVID-19 and to predict the epidemic spread analysis of the risk in a region at the beginning of the outbreak [21]. Dandekar et al. used mixed first-principles epidemiological equations and data-driven neural network models to interpret and extrapolate from publicly available data the effect of quarantine interventions to control the epidemic in all the stages of the outbreak [28].

About the outcomes, as it has been shown in Table 1, four models estimated the probability that a SARS-COV-2 outbreak could be controlled [22,23,24,28]. In the study performed by Wang et al., the outcomes included the reduction in the epidemic spike and the probability to avoid the second wave of the infection [27]. The other studies adopted outcomes such as forecasting the risk of new hotspot formation [21], the prediction of the outbreak evolution and the rate of recovery [25], and the reduction in the transmission rate [26].

In Table 2, the effectiveness of the interventions (single and multiple interventions) is described. In fact, interventions such as a stringent quarantine and a massive lockdown significantly reduced the transmission rate of COVID-19 and avoided more than 1.4 million infections and 56,000 deaths in China [26]. Another study showed that a partial lockdown had a strong impact on eventual infection fraction (*x*^~^(adjusted-R2 = 0.59, *p* = 2 × 10^−6^)) and concluded that the lockdown should be implemented before the peak infection [27]. The models that used empirical data showed that quarantine was effective in controlling the epidemic spread also as a part of single and multiple interventions in all the stages of the outbreak [26,27,28].

Social distancing policies that were implemented in 26 countries showed a reduction in disease transmission rates (47% variation) and were effective in flattening the curve [22].

The models that used AI and ML to simulate the effectiveness of intervention showed similar results. A mathematical and ML modeling study that simulated an intervention with lockdown measures concluded that the lockdown with certain restrictions might help in preventing the spread of the epidemic [24].

A model that simulated multiple interventions such as the closure of schools and activities, the limitation of public transport and the adoption of social distancing showed that without such interventions, 42.7% of the Swiss population would have been infected by 25 April 2020 compared with the observed 1% infection rate over the period [25].

A different approach was adopted by Kumar et al. and evaluated the effectiveness of lockdown according to the level of diffusion of the virus. In low-risk areas, the study showed that releasing all constraints except mass gatherings and traveling out of the district should be effective. In moderate-risk areas, releasing partial constraints except mass gatherings and travel out of the district and markets with essential commodities should be effective. In high-risk areas, lockdown should be increased, sealing the districts with essential commodities at doorsteps in order to be effective [21].

The adoption of quarantine of the people with infectious status (I-status) and reducing their movement could be effective in controlling the spread of the epidemic. This study also recommended that if medical resources are available, the exposed status (E-status) individuals or potential E-status individuals should be included in the scope of isolation and treatment. Moreover, the government should promptly release information on the epidemic situation and information on the areas and vehicles used by the infected people to further encourage those who have been in contact with individuals (I-status or E-status) to go to nearby hospitals for immediate inspection [23].

## 4. Discussion

Based on our findings, quarantine emerged as the most effective intervention to control the spread of COVID-19 [23,26,28]. China implemented a combination of interventions based on quarantine that also included the implementation of cordon sanitary measures and traffic restrictions from 23 January 2020 to 16 February 2020. Before the implementation, the Rt was above 3.0. After the application of the quarantine, on 6 February 2020 the Rt decreased to below 1.0, and on 1 March 2020 the Rt decreased to less than 0.3 [29]. The data of 190 countries worldwide that implemented the quarantine measures (from 23 January 2020 to 13 April 2020) showed how they were associated with a reduction in Rt when compared with countries that did not adopt this measure (Rt = −11.40%, 95% CI (−9.07–−13.66%)) [30].

AI and ML were also applied in the use of lockdown [26]. The results of eleven European countries that implemented a lockdown between 3 February 2020 and 4 May 2020 showed a reduction in Rt below 1 and a large effect on reducing transmission [31]. A recent study that ranked the effectiveness of worldwide COVID-19 public health interventions that were implemented in 79 territories showed that curfews, cancellations of small gatherings and closures of schools, shop and restaurants were among the effective public health policies [32]. All these results were consistent with the outputs of the quarantine and lockdown-based AI and ML models [23,26,27,28].

AI and ML also simulated the adoption of continuously redefining the modification of lockdown measures according to the spatial (area) risk of the spread of the disease in one area (low, moderate, and high) [21]. This intervention was mainly used by Western European countries. Additionally, India implemented the same approach during lockdown phase 3 (from 4 May 2020 to 17 May 2020). After the application of this measure, the Rt decreased from 2.78 to 1.38. In brief, even though this approach reduced the spread of COVID-19 epidemic progression, it was unable to halt and eventually eradicate the COVID-19 epidemic [33].

Social distancing was the last strategy that was evaluated with AI and ML. AI and ML suggested that social distancing could be effective only in combination with the closure of schools/commercial activities and the limitation of public transportation [25]. Additionally, from real life data the application of social distancing as a single intervention was not very successful because case resurgence was likely to occur once it was removed and it did not help to reduce the excess mortality [34,35].

The models used in our study are quite diverse and a few considerations about their characteristics are worthwhile. The main models considered are the following: SIR/SEIR (Susceptible–Exposed–Infected–Recovered), Linear Regression, TOPSIS, Neural Networks, Agent-based Simulation.

These models are from very different families of methods, ranging from differential equation models (SIR/SEIR), to statistical machine learning models (linear regression and neural nets), geometric models (TOPSIS), and, finally, simulation models (agent-based simulation). A direct comparison is then hard, and the choice of one method with respect to another one may depend upon several factors, such as the kind of collected data, the availability of analytical tools, and the contextual situation under which the model can actually be applied. For instance, the SIR family of models, as any model in system theory (i.e., differential equations), assumes that the modeled system abstracts to some specific behavior. In particular, in standard SIR, a homogeneous mixing of the infected *I* and susceptible *S* populations is assumed, meaning that a person’s contacts are randomly distributed among all others in the population. However, in real situations, the mixing in a population is heterogeneous and contacts are usually not random; for example, people of different ages may have very different kinds of relationships.

Machine learning models do not assume such a kind of abstract behavior, since they try to predict specific patterns of prediction from data; in other words, they tend to learn the abstract behavior of the system from observations, and they use what has been learnt to make predictions. However, in this case specific modeling assumptions are also present. Standard linear regression is a model with very high bias, since it assumes a linear relationship between observed data and the target; however, the bias can be reduced by adjusting the model to polynomial regression with the introduction of additional non-linear (quadratic, cubic, etc.) parameters. It is well-known that this bias reduction will increase the variance of the model, leading to the problem of overfitting (the inability of the model to generalize to unobserved data, while being really accurate on observed data). Regularization techniques (lasso or L2 regularization) can be adopted to reduce overfitting [36]. Neural networks are more general, since the non-linearity can be captured in the activation functions of the artificial neurons (usually sigmoid functions such as logistic or hyperbolic tangent, as well as Rectified Linear Unit widely adopted in deep neural net modes), and overfitting can be mitigated by both suitable architectural choices as well as regularization. However, the choice of the right set of hyper-parameters of the net (number of neurons, number of hidden layers, activation functions) and of the learning algorithm (learning rate, momentum, parameter initialization) may have a great impact on the final model’s performance and must be made by intensive cross-validation procedures.

Geometric models such as TOPSIS more directly address a decision-making process and are quite interesting in a setting like the one discussed in the present paper, i.e., the evaluation of specific countermeasures to contain the spread of COVID-19. In particular, TOPSIS belongs to the class of Multiple Attribute Decision Making (MADM) approaches, where some courses of action are chosen in the presence of multiple, usually conflicting, features. An interesting observation is that similar approaches have also been investigated in the Machine Learning community with the use of Probabilistic Graphical Models, such as Decision Networks or Influence Diagrams [37], but with the possibility of learning both the structural relationship among the attributes and their quantification in terms of uncertainty (probability) and utility.

Finally, agent-based simulation is a completely different alternative, where no specific modeling is assumed, but the results are obtained by looking at the interactions among the involved agents. The crucial point is to determine the right set of simulation parameters, such as the number of agents, the rate of interaction, the probability of infection given by contacts, etc.

In summary, all the approaches investigated in the different studies have their motivations, as well as their strengths and limitations, and no one can be, in general, considered better or worse than another one. However, the finding suggesting that quarantine is a good and efficient strategy for containing COVID-19 is an important result which is strengthened by the convergence of such different models.

## 5. Limitations of the Study

Our study has limitations too. First, we did not have the possibility to use the risk of bias assessment tool, since no validated bias checklist is available. A few studies that are included in our review are still in the preprint stage [4,21,23,24,25,27,28,34]. We draw conclusions from a few studies (*n* = 8). Additionally, some studies analyzed interventions in one single country. Therefore, we cannot conclude that they can also be efficient in other countries [21,23,24,25,26,27]. It was also difficult to distinguish the consequences of a single policy measure from those of other policy interventions. Although there were a variety of mathematical methods for unravelling relationships in structural components, none of them were ideal. Lastly, all the studies were performed at the beginning of the pandemic before the emergence of COVID-19 variants and before the introduction of the vaccines. This could be a major limitation to the actual use of our findings because the new COVID-19 variants could have different transmission patterns and the national vaccination program will substantially change the effects of interventions over time [38].

## 6. Conclusions

Despite the possible limitations, the outputs of AI and ML were generally consistent with the results obtained by most of the public health interventions that have been used to reduce the spread of COVID-19 worldwide. Our study findings showed that AI and ML could have been useful to help policy makers to better define the best strategies for containing the COVID-19 pandemic since the end of the first wave. As a matter of fact, at least half the articles (four of the seven for which dates could be clearly identified) were published in April 2021 or later—with the last two being “published” in May and June 2021, respectively. In particular, quarantine clearly emerged as the best strategy for containing COVID-19. On the contrary, a strict quarantine was rarely adopted worldwide. Additionally, according to AI and ML outputs, total, early and time extensive nationwide lockdown should have been adopted to stop the second wave because of the effectiveness in reducing the Rt and the transmission of the disease. On the contrary, such a measure was rarely adopted fully and often has been continuously mitigated according to the variations in the local risk of the spread of the disease. In fact, even though this strategy could not stop the pandemic, it was probably more acceptable because it did not drastically affect the people’s degree of freedom due to the pandemic [39]. Social distancing should have been considered effective only with a combination of interventions, but again it was also widely adopted as a single intervention. We believe that this happened because most of the public interventions for preventing the second wave were implemented based mainly on their mechanistic or biological plausibility. On the contrary, they could have taken advantage of the use of AI and ML outputs. Even though the time lag between when the decisions needed to be made and these data appeared to be ready to be used makes our finding more effective as a summative evaluation than as a process evaluation, we still think that, in the near future, Al and ML should play a significant role in public health policy making.

## Figures and Tables

**Figure 1 ijerph-18-04499-f001:**
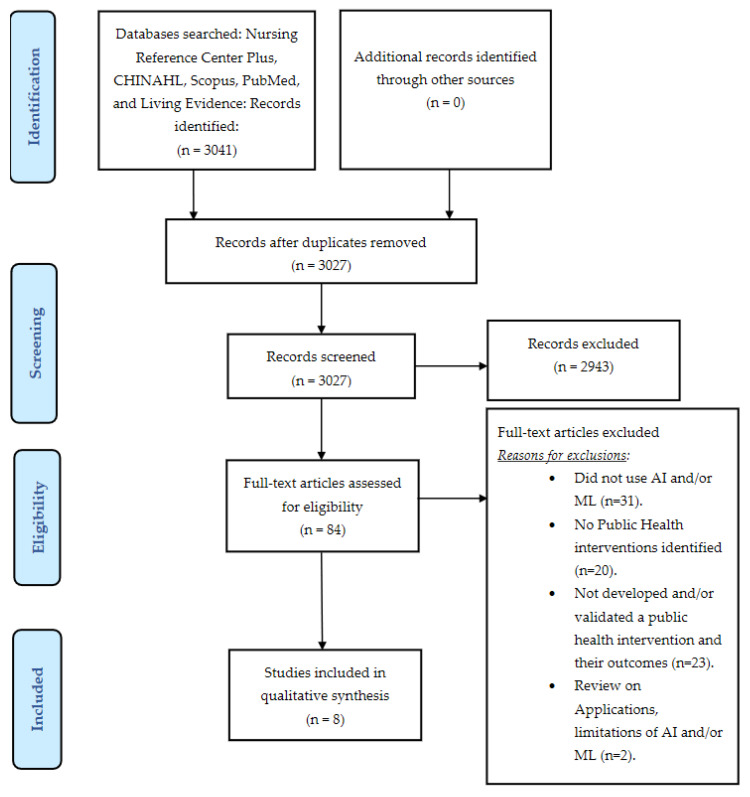
PRISMA Flow Diagram for the selection of articles.

**Table 1 ijerph-18-04499-t001:** Types of models and typology of data, and their setting in the included studies.

Title	Author	Setting	Outcome	Model Development	Model Characteristic	Typology of Data
On the Spread of Coronavirus Infection. A Mechanistic Model to Rate Strategies for Disease Management.	Shiyan Wang	United States	Control of the epidemic spread, reduce spike.	New	Mechanistic	Empirical
No Place Like Home: Cross-National Data Analysis of the Efficacy of Social Distancing During the COVID-19 Pandemic.	Dursun Delen	26 countries	Control of the epidemic spread, reduce spike.	Existing	Susceptible–infected–recovered (SIR)	Empirical
Predicting the COVID-19 positive cases in India with concern to Lockdown by using Mathematical and Machine Learning based models.	Ajit Kumar Pasayat	India	Control of the epidemic spread, reduce spike.	Existing	Exponential Growth, Linear Regression	Simulation
Preparedness and Mitigation by projecting the risk against COVID-19 transmission using Machine Learning Techniques.	Akshay Kumar	India	Risk of hotspot formation.	New	Technique for Order of Preference by Similarity to Ideal Solution (TOPSIS)	Simulation
Quantifying the effect of quarantine control in COVID-19 infectious spread using machine learning.	Raj Dandekar	Wuhan, Italy, South Korea, USA	Control of the epidemic spread.	New	Neural network augmented	Empirical
COVID-19 Epidemic in Switzerland: Growth Prediction and Containment Strategy Using Artificial Intelligence and Big Data.	Marcello Marini	Switzerland	Outbreak prediction evolution of spread, rate of recovery.	Existing	Agent-based simulation framework, EnerPol	Simulation
Impacts of Social and Economic Factors on the Transmission of Coronavirus Disease 2019 (COVID-19) in China.	Yun Qiuy	China	Reduce the transmission rate.	Existing	Empirical	Empirical
Beware of asymptomatic transmission: Study on 2019-nCoV prevention and control measures based on extended SEIR model.	Peng Shao	China	Control of the epidemic spread.	Existing	Susceptible–Exposed–Infectious–Recovered (SEIR)	Simulation

**Table 2 ijerph-18-04499-t002:** Effectiveness of the interventions.

Author	Outbreak Phase	Intervention Type	Description of Intervention	Results
Shiyan Wang	All the stages of the epidemic	Multiple	(i.) Stay at home order. (ii.) Easing social distancing measures. (iii.) Mandatory quarantine for travelers. (iv.) Non-essential business closure. (v.) Gathering ban. (vi.) School closure. (vii.) Restaurant limits.	The study suggested that non-essential business closure, a gathering ban and school closure could have a strong impact on eventual infection fraction—if the interventions were implemented before the peak infection rate.
Dursun Delen	All the stages of the epidemic	Single	Social Distancing.	Social distancing policies could help in slowing the spread of COVID-19 (approximately 47% of the variation in the disease transmission rates) as well as in flattening the epidemic curve.
Ajit Kumar Pasayat	All the stages of the epidemic	Single	(i.) Lockdown is not continuing strictly after May 18th, 2020. (ii.) Lockdown continues.	Partial lockdown could play a positive role in preventing the spread of the disease.
Akshay Kumar	Beginning of the epidemic	Single	Adaption of lockdown measures according to the risk (low, moderate, and high) of new hot spots.	The study suggested to: (i) Release all constraints except mass gatherings and travel out of district in low-risk areas. (ii) Release partial constraints, i.e., (i) + markets with essential commodities in moderate-risk areas. (iii) Seal the districts with essential commodities at doorsteps in high-risk areas.
Raj Dandekar	All the stages of the epidemic	Single	Quarantine and isolation.	Strong correlation between strengthening of the quarantine, actions taken by governments, and a decrease in effective reproductive number (Rt).
Marcello Marini	Beginning of the epidemic	Multiple	(i.) Closure of schools. (ii.) Closure of activities. (iii.) Limitation of public transport. (iv.) Social distancing.	The study estimated that, in the absence of interventions, 42.7% of the Swiss population would have been infected.
Yun Qiuy	Beginning of the epidemic	Multiple	(i.) Stringent quarantine. (ii.) Massive lockdown. (iii.) Other public health measures.	The interventions significantly reduced the transmission rate of COVID-19. The study also demonstrated that the actual population flow from the outbreak source poses a higher risk to the destination than geographic proximity and similarity in economic conditions.
Peng Shao	Beginning of the epidemic	Multiple	(i.) Quarantine of infected people. (ii.) Reduction in movement of people.	The measures could help in controlling the spread of the epidemic.

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
