# Peer review of "Prediction Models for Public Health Containment Measures on COVID-19 Using Artificial Intelligence and Machine Learning: A Systematic Review"

_ijerph, 2021, doi:10.3390/ijerph18094499_

Round 1
Reviewer 1 Report
I have two major concerns about this paper.
The first is that it is seeking to base policy decisions on limited input. You – as the authors – even noted the few studies as a limitation of this research. Drawing conclusion from such a small sample size is fraught with peril. (It also would have been instructive to know why so many of those originally selected articles were discarded.)
Even more problematic is the time lag. At least half the articles (four of the seven for which dates could be identified) were published in April or later – with the last two being “published” in May and June respectively. By that point, decisions already had to have been made – right or wrong – to contain the pandemic.
Furthermore, it is now March 2021 – 12 to 15 into the pandemic, depending upon where you are in the world. The time lag between when the decisions needed to be made and this data appears to be ready to be used makes it apparent that this systematic review of artificial intelligence and machine learning is more effective as a summative evaluation than as a process evaluation. In other words, it can teach us how to act in the future, if (when) a similar situation arises. But does not appear to have much to offer the present situation as other factors (such as vaccines) have changed the situation substantially since these initial reports on mitigation activity.
In other words, is the conclusion accurate? Can you really use this to help decision-makers, as it is March 2021 – a year (more or less) since these decisions had to be made (Line 246)? Or is it more an evaluation tool and examples for future situations?
I also had some other structural comments.
- Did both reviewers extract from all articles or did each do half the articles? It appears the former, but that should be explicit (Line 67).)
- The authors need to only be listed in the text by last name. Many are listed by last name followed by first initial or even full first name (multiple times, beginning on Line 102).
- The word “sanitaire” does not make sense; is it supposed to be “sanitary” (Line 203)?
- The date format needs to be standardized throughout. Sometimes dates are listed with ordinal numbers and other times with cardinal numbers. Likewise, some are listed in Day Month Year” format while others are listed in “Month Day, Year” format. (And one date is only Month Year (Line 214)).
- Use full words – did not – rather than contractions (“didn’t” on Line 234).
Reviewer 2 Report
In the Reviewer opinion the research paper entitled “Prediction models for public health containment measures on COVID-19 using Artificial intelligence and Machine learning: A systematic review” is good.
This systematic review is aimed to evaluate the effectiveness of the use of AI and ML when applied to public health interventions to contain the widespread of SARS-CoV-2. Our findings showed that quarantine should be the best strategy for containing COVID-19. Nationwide lock-down also showed positive impact, whereas social distancing should be considered to be effective only in combination with other interventions including closure of school and commercial activities, and limitation of public transportation. This findings also showed that all the interventions should be initiated early in the pandemic and continued for a sustained period. Despite the study limitation, authors conclude that AI and ML could be of help for policymakers to define the strategies for containing the COVID-19 pandemic.
Some comments which greatly enhance the understanding of the paper and its value are presented below. Specific issues that require further consideration are:
- The title of the manuscript is matched to its content.
- In the Reviewer’s opinion, the current state of knowledge relating to the manuscript topic has been more-less presented, but the author's contribution and novelty are not enough emphasized.
- In the Reviewer’s opinion, the bibliography, comprising 36 references, is rather representative.
- An analysis of the manuscript content and the References shows that the manuscript under review constitutes a summary of the Author(s) achievements in the field. However, the introduction needs more attention.
- Conclusion needs to be more revised and extended.
- In the Reviewer’s opinion the manuscript should be published in the journal after major revision.
Reviewer 3 Report
This article presents a systematic review and summarisation of studies that make use of statistical, AI and ML models to predict the effectiveness of public help interventions in the context of the ongoing COVID-19 pandemic. The authors perform automated literature extraction from a number of bibliographical databases with the help of keywords, followed by a manual filtering of eligible studies, ending with 8 candidates. These candidates represent an interesting sample of heterogeneous models, methodologies and empirical settings. Quarantine was found to be the most prevalent policy that was found to be effective across these studies, followed by nationwide lockdown.
The procedure to obtain the relevant studies appears to be rigorous and sound.
I must make it clear that I am a Computer Scientist, specialised in Artificial Intelligence and Machine Learning. I am used to interdisciplinary research, but I have no expertise in Public Health or Epidemiology. In the topic being addressed, I believe that interdisciplinary research is indeed of the utmost importance.
My overall impression from this work is that it follows a process of comparison that is conventional in the medical field (e.g. meta-analysis to access the effectiveness of a drug), but less usual in comparing AI/ML approaches. I find that there are two main issues:
(1) The models uses in the 8 studies are indeed diverse, including SIR, linear regression, neural networks and simulation / agent-based modelling. These various approaches have specific strengths and weaknesses, as well as different assumptions and simplifications, and the direct comparison of their predictions is far from trivial. The article does not discuss such issues, making it very hard to evaluate the strength of the conclusions.
(2) A high-level comparison of the conclusions of the studies with the actual known outcomes of public policies is presented, but I find this quite unsatisfactory. The various models propose to predict outcomes under certain scenarios, and this article would be vastly strengthened if it also discussed the success of each model in making quantitative predictions against emerging data, not to mention confidence intervals and sensitivity analysis. I understand this is very hard, both because the situation is still unfolding and also because the credibility of available data varies widely and is still uncertain.
While I appreciate the literature search and filtering, I fear that in the current state this article runs the risk of comparing "apples to oranges", and not provide enough context nor quantitative information to allow the reader to emerge with a correct understanding of the findings.
I would recommend the inclusion of a AI/ML expert to extend the work with an appropriate discussion of the strengths and weaknesses of the various models, as well as a proper evaluation of the quality of their predictions.
I regret saying that I am of the opinion that this work should not be published in the absence of such a major revision.
Round 2
Reviewer 1 Report
You have addressed my comments and improved the paper tediously.
From a read of your response letter, if appears we both had the same idea regarding the utility of the study. That comes through much more clearly now than it did in the original version of the paper.
The clarifying language helps to show the intent of this study – something which I found a bit more elusive to find previously. Likewise, the added details explain anticipated questions that the reader may have as well as expand on the rationale for undertaking this study.
At this point, the subject matter is fine. However, there are still a few matters that need to be addressed with respect to the presentation. The most prominent problem are the incomplete sentences used to describe the application of AI and ML to public health issues related to COVID-19. The three sentences from Line 37-40 should be combined into a single sentence separated with semi-colons (or verbs need to be added to the second and third sentence). Likewise, the added discussion about optimizing protection in Line 40-44 lacks a predicate (verb).
[There is also inconsistent paragraph indenting in Section 3. There is extra spaces in the indenting at the beginning of the paragraph on Line 96 and an extra tab space at the beginning of the paragraph on Line 103. However, this would be corrected in typesetting.]
Once the paper has received a final proofreading – and correction of any issues found – it should be (in my opinion) ready for publication.
Reviewer 2 Report
Authors corrected article and it can be published in the Journal.
Author Response
Dear Reviewer,
We appreciate the time and effort you put into providing feedback on our manuscript. Thank you for your acceptance of the manuscript for publication.